# Integrative analysis of large-scale loss-of-function screens identifies robust cancer-associated genetic interactions

**Christopher J Lord[1], Niall Quinn[2], Colm J Ryan[2]***

[1]Breast Cancer Now Toby Robins Research Centre and Cancer Research UK Gene Function Laboratory, Institute of Cancer Research, London, United Kingdom; [2]School of Computer Science and Systems Biology Ireland, University College Dublin, Dublin, Ireland

**Abstract** Genetic interactions, including synthetic lethal effects, can now be systematically identified in cancer cell lines using high-throughput genetic perturbation screens. Despite this advance, few genetic interactions have been reproduced across multiple studies and many appear highly context-specific. Here, by developing a new computational approach, we identified 220 robust driver-gene associated genetic interactions that can be reproduced across independent experiments and across non-overlapping cell line panels. Analysis of these interactions demonstrated that: (i) oncogene addiction effects are more robust than oncogene-related synthetic lethal effects; and (ii) robust genetic interactions are enriched among gene pairs whose protein products physically interact. Exploiting the latter observation, we used a protein–protein interaction network to identify robust synthetic lethal effects associated with passenger gene alterations and validated two new synthetic lethal effects. Our results suggest that protein–protein interaction networks can be used to prioritise therapeutic targets that will be more robust to tumour heterogeneity.

*For correspondence:
colm.ryan@ucd.ie

**Competing interests:** The authors declare that no competing interests exist.

## Introduction

Large-scale tumour genome sequencing efforts have provided us with a compendium of driver genes that are recurrently altered in human cancers (*Vogelstein et al., 2013*). In some cases, these genetic alterations have been associated with altered sensitivity to targeted therapies. Examples of targeted therapies already in clinical use include approaches that exploit oncogene addictions, such as the increased sensitivity of *BRAF* mutant melanomas to BRAF inhibitors (*Chapman et al., 2011*), and approaches that exploit non-oncogene addiction/synthetic lethality, such as the sensitivity of *BRCA1/2* mutant ovarian or breast cancers to PARP inhibitors (*Lord and Ashworth, 2017*). An ongoing challenge is to associate the presence of other driver gene alterations with sensitivity to existing therapeutic agents (*Barretina et al., 2012*; *Iorio et al., 2016*) or to identify candidate therapeutic targets whose inhibition may provide therapeutic benefit to patients with specific mutations. Towards this end, multiple groups have performed large-scale loss-of-function genetic perturbation screens in panels of tumour cell lines to identify vulnerabilities that are associated with the presence or absence of specific driver gene mutations (i.e. genetic interactions) (*Behan et al., 2019*; *Campbell et al., 2016*; *Marcotte et al., 2016*; *McDonald et al., 2017*; *Meyers et al., 2017*; *Tsherniak et al., 2017*). Others have performed screens in 'isogenic' cell line pairs that differ only by the presence of a specific oncogenic alteration (*Martin et al., 2017*; *Steckel et al., 2012*). Despite these large-scale efforts, very few genetic interactions have been identified in more than one study (recently reviewed *Ryan et al., 2018*). Even in the case of cancer driver genes subjected to multiple screens, such as *KRAS*, few genetic interactions have been identified in more than one

screen (*Downward, 2015*). This lack of reproducibility may be due to technical issues, for example false positives and false negatives due to inefficient gene targeting reagents (*Kaelin, 2012*), and/or real biological issues, such as the context specificity of genetic interactions (*Henkel et al., 2019*; *Ryan et al., 2018*). We refer to those genetic interactions that can be reproduced across multiple screens and across distinct cell line contexts as *robust genetic interactions*. Given that tumours exhibit considerable molecular heterogeneity both within and between patients there is a real need to: (i) identify robust genetic interactions that can be reproduced across heterogeneous cell line panels, reasoning that these reproducible effects will be more likely to be robust in the face of the molecular heterogeneity seen in human cancers; (ii) prioritise these robust genetic interactions for further therapeutic development; and (iii) understand the characteristics of robust genetic interactions in cancer as a means to predict new therapeutic targets.

To achieve this, we developed, and describe here, a computational approach that leverages large-scale cell line panel screens to identify those genetic interactions that can be reproducibly identified across multiple independent experiments. We found that for all oncogenes studied, the most significant reproducible dependency identified was an oncogene-addiction rather than a synthetic lethal effect. Excluding oncogene addictions, we found 220 reproducible genetic interactions. In investigating the nature of these robust genetic interactions, we found that they are significantly enriched among gene pairs whose protein products physically interact. This suggests that incorporating prior knowledge of protein–protein interactions may be a useful approach to guide the selection of reproducible 'hits' from genetic screens as candidates worth considering as therapeutic targets in cancer. We demonstrate the utility of the approach in identifying robust synthetic lethal interactions from chemogenetic screens and in identifying synthetic lethal interactions associated with 'passenger' gene alterations.

## Results

### A 'discovery and validation' approach to the analysis of loss-of-function screens identifies reproducible genetic dependencies

We first wished to identify genetic interactions that could be independently reproduced across multiple distinct loss-of-function screens. To do this, we obtained gene sensitivity scores from four large-scale loss-of-function screens in panels of tumour cell lines, including two shRNA screens (DRIVE [*McDonald et al., 2017*], DEPMAP [*Tsherniak et al., 2017*]) and two CRISPR-Cas9 mutagenesis screens (AVANA [*Meyers et al., 2017*], SCORE [*Behan et al., 2019*]). We harmonised the cell line names across all studies, so they could be compared with each other and also with genotypic data (*Barretina et al., 2012*; *Iorio et al., 2016*; *Figure 1A*). In total, 917 tumour cell lines were screened in at least one loss-of-function study. Only 50 of these cell lines were common to all four studies while 407 cell lines were included in only a single study (*Figure 1B*). It is the partially overlapping nature of the screens that motivated the subsequent approach we took for our analysis. We used a 'discovery set' and 'validation set' approach to identifying genetic interactions across multiple screens - first identifying associations between driver gene alterations and gene inhibition sensitivity in the discovery study and then testing the discovered dependency in the validation study (*Figure 1C*). However, to ensure that any reproducibility observed was not merely due to cell lines common to both datasets, we first removed cell lines from the validation dataset if they were present in the discovery dataset (*Figure 1C*). For example, when using DEPMAP as the discovery dataset and AVANA as the validation dataset, we performed the validation analysis on the subset of cell lines that were present in AVANA but not in DEPMAP. In doing so, we ensured that any genetic interactions discovered were reproducible across different screening platforms (either distinct gene inhibition approaches, that is shRNA vs CRISPR, or distinct shRNA/CRISPR libraries) and also robust to the molecular heterogeneity seen across different cell line panels.

Similar to our previous work (*Bridgett et al., 2017*; *Campbell et al., 2016*) we integrated copy number profiles and exome sequencing data to annotate all cell lines according to whether or not they featured likely functional alterations in any one of a panel of cancer driver genes (*Vogelstein et al., 2013*) (see Materials and methods, *Supplementary file 1*). We then identified associations between driver gene alterations and sensitivity to the inhibition of specific genes using a multiple regression model. This model included tissue type, microsatellite instability and

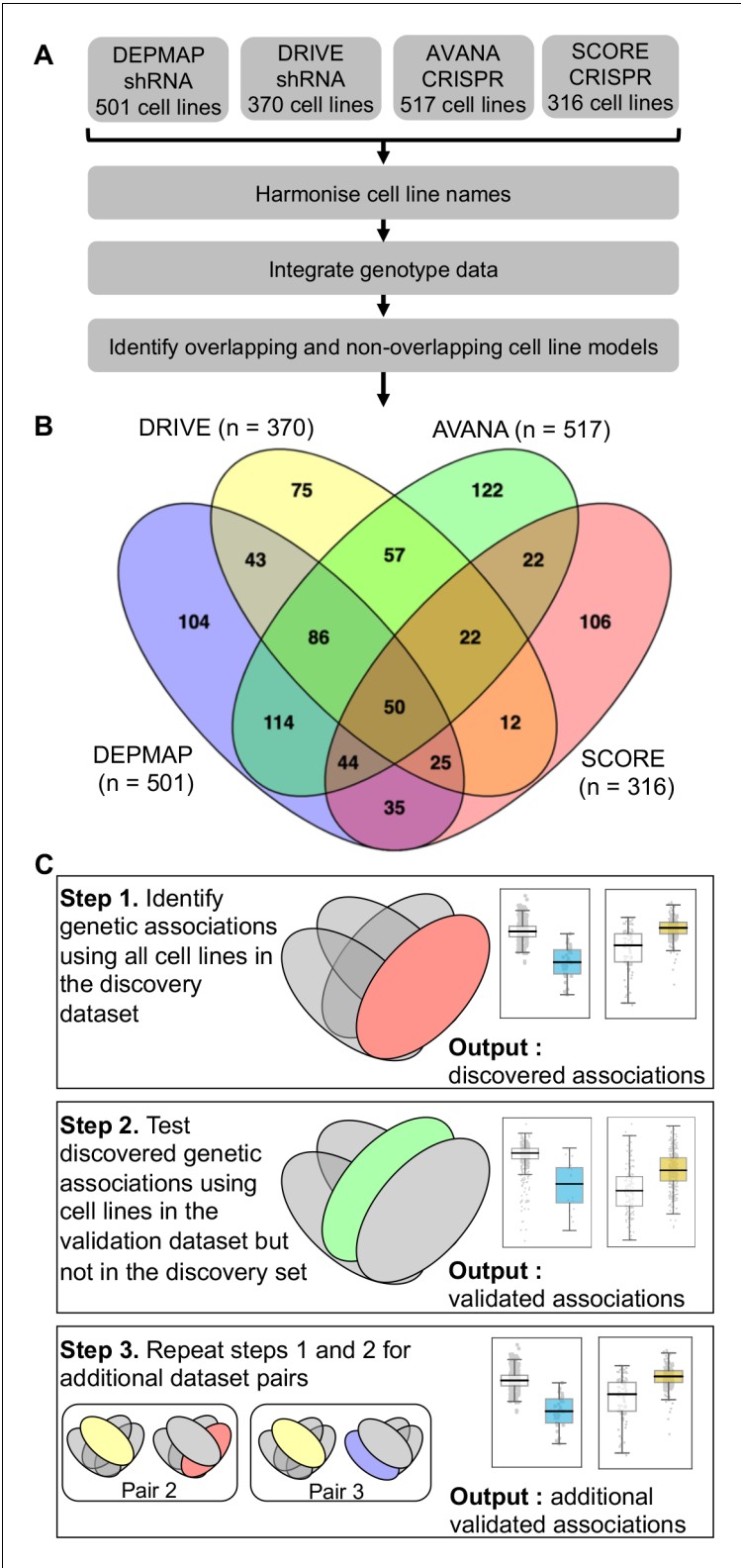

**Figure 1.** Identifying robust genetic interactions using partially overlapping loss-of-function screens. (**A**) Workflow showing the integration of four different loss-of-function screen datasets. (**B**) Venn diagram showing the overlap of cell lines between the four datasets analysed in this study. (**C**) Workflow showing how robust genetic interactions are identified using discovery and validation sets.

driver gene status as independent variables and gene sensitivity score as the dependent variable (Materials and methods). Microsatellite instability was included as a covariate as it has previously been shown to be associated with non-driver gene specific dependencies (*Behan et al., 2019*), while tissue type was included to avoid confounding by tissue type. We focused this analysis on 'selectively lethal' genes - that is those genes whose inhibition killed some, but not all cell lines (Materials and methods, *Supplementary file 2*). We analysed each pair of screens in turn and considered a genetic dependency to be reproducible if it was validated in at least one discovery/validation pair. Using this approach, we tested 142,477 potential genetic dependencies between 61 driver genes and 2421 selectively lethal genes. We identified 1530 dependencies that were significant in at least one discovery screen (*Figure 2A*, *Figure 2—figure supplement 1*). All 61 driver genes had at least one dependency that was significant in at least one discovery screen while less than half of the selectively lethal genes (1,141/2,421) had a significant association with a driver gene. Of the 1530 dependencies that were significant in at least one discovery screen, only 229 could be validated in a second screen (*Supplementary file 3*, *Figure 2A*). For example, in the AVANA dataset *TP53* mutation was associated with resistance to inhibition of both *MDM4* and *CENPF*, but only the association with *MDM4* could be validated in a second dataset (*Figure 2B and C*). Similarly, in the DEPMAP dataset *NRAS* mutation was associated with increased sensitivity to the inhibition of both *NRAS* itself and *ERP44*, but only the sensitivity to inhibition of *NRAS* could be validated in a second dataset (*Figure 2B and C*).

The 229 reproducible dependencies involved 31 driver genes and 204 selectively lethal genes. Of the 229 reproducible genetic dependencies nine were 'self *vs.* self' associations, where the alteration of a gene was associated with sensitivity to its own inhibition. The majority (7/9) of these 'self *vs.* self' associations were oncogene addiction effects, such as the increased sensitivity of *NRAS* mutant cell lines to *NRAS* inhibition (*Figure 2B*). Similarly, we identified robust oncogene addictions involving the *CTNNB1* (β-Catenin), *KRAS*, *EGFR*, *BRAF*, *ERBB2* and *PIK3CA* oncogenes (*Figure 3A and B*, *Figure 3—figure supplement 1B*). For *EGFR* and *CTNNB1*, the only identified robust dependency was an oncogene addiction effect. For all other oncogenes there were additional robust dependencies identified, but in all cases the most significant reproducible dependency was an oncogene addiction (*Figure 3—figure supplement 1A*). These observations suggest that for most oncogenes the oncogene addiction effect might be more robust than any oncogene-related synthetic lethal effects.

We also identified two (2/9) examples of 'self *vs.* self' dependencies involving tumour suppressors -*TP53* (aka p53) and *CDKN2A* (aka p16/p14arf) (*Figure 3—figure supplement 1C*). This type of relationship has previously been reported for *TP53*: *TP53* inhibition appears to offer a growth advantage to *TP53* wild type cells but not to *TP53* mutant cells (*Giacomelli et al., 2018*). Inhibiting *TP53* in *TP53* mutant cells has a largely neutral effect, while on average inhibiting *TP53* in *TP53* wild type cells actually increases fitness growth. Consequently, we observed an association between *TP53* status and sensitivity to TP53 inhibition. Similar effects were seen for *CDKN2A,* although the growth increase resulting from inhibiting *CDKN2A* in wild-type cells is much lower than that seen for *TP53* (*Figure 3—figure supplement 1C*). These 'self *vs.* self' dependencies, in particular the oncogene addictions, serve as evidence that our approach could identify well characterised genetic associations. However, as our primary interest was in genetic interactions between different genes, we excluded 'self *vs.* self' interactions from further analysis, leaving us with 220 robust genetic interactions (*Figure 3C*).

## Many robust genetic interactions reflect known pathway structure

Many of the robust genetic interactions we identified have been previously reported, including both sensitivity relationships, such as increased sensitivity of *PTEN* mutant cell lines to inhibition of the phosphoinositide 3-kinase-coding gene *PIK3CB* (*Wee et al., 2008*), and resistance relationships, such as an increased resistance of *TP53* mutant cell lines to *MDM2* inhibition (*Figure 3D*).

Amongst the set of 220 robust genetic interactions, we identified two previously reported 'paralog lethalities' – synthetic lethal relationships between duplicate genes (paralogs) (*Helming et al., 2014*; *Hoffman et al., 2014*; *Oike et al., 2013*; *Figure 3E*). We found a robust association between mutation of the tumour suppressor *ARID1A* and sensitivity to inhibition of its paralog *ARID1B* (*Helming et al., 2014*) and also an association between mutation of *SMARCA4* and sensitivity to inhibition of its paralog *SMARCA2* (*Hoffman et al., 2014*; *Oike et al., 2013*). Both pairs

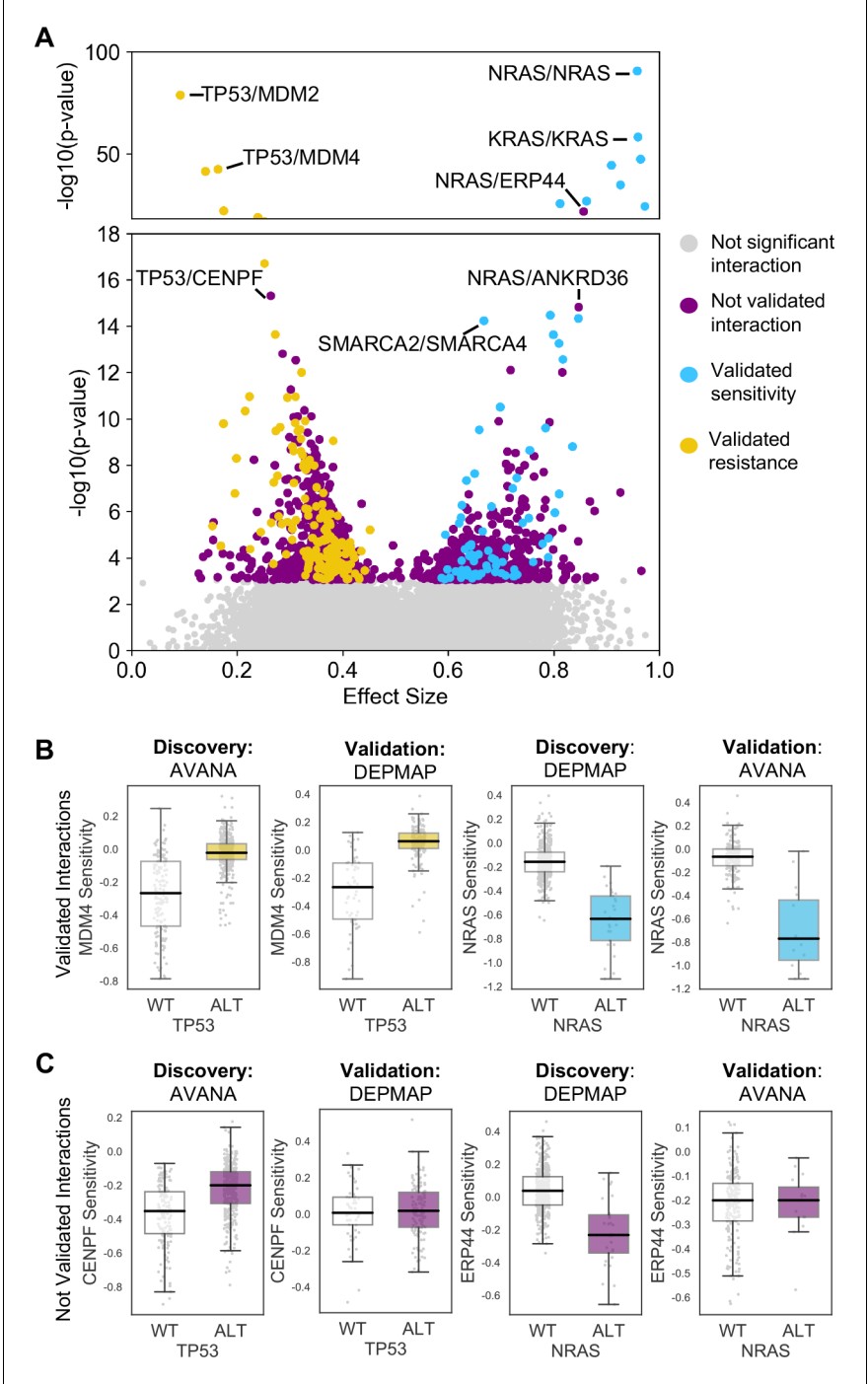

**Figure 2.** Discovered and validated genetic dependencies. (**A**) Scatterplot showing the genetic dependencies identified across all datasets. Each individual point represents a gene pair, the x-axis shows the common language effect size, and the y-axis shows the -log10 p-value from the discovery dataset. Selected gene pairs are highlighted – the driver gene is listed first, followed by the associated dependency. Each gene pair may have been tested in multiple discovery studies, only the interaction with the most significant discovery p-value is shown. Scatterplots for individual studies are presented in *Figure 2—figure supplement 1*. (**B**) Tukey boxplots showing examples of robust genetic dependencies, including an increased resistance of *TP53* mutant tumour cell lines to *MDM4* inhibition and increased sensitivity of *NRAS* mutant tumour cell lines to *NRAS* inhibition. In each box plot the top and bottom of the box represents the third and first quartiles and the box band represents the median; whiskers extend to 1.5 times the interquartile distance from the box. WT = wild type, ALT = altered. Throughout blue is used to indicate increased sensitivity (synthetic lethality or oncogene addiction), yellow to indicate

*Figure 2 continued on next page*

*Figure 2 continued*

resistance to inhibition of the target gene. (**C**) Boxplots showing examples of genetic dependencies discovered but not validated, including an increased resistance of *TP53* mutant cell lines to *CENPF* inhibition and increased sensitivity of *NRAS* mutant cell lines to *ERP44* inhibition.

The online version of this article includes the following figure supplement(s) for figure 2:

**Figure supplement 1.** Discovered and validated genetic dependencies for individual datasets.

---

of genes (*ARID1A/ARID1B* and *SMARCA4/SMARCA2*) encode components of the larger SWI/SNF complex (*Wilson and Roberts, 2011*).

Some of the robust genetic dependencies could be readily interpreted using known pathway structures. For instance, many of the robust dependencies associated with the oncogene *BRAF* could be interpreted in terms of BRAF's role in the MAPK pathway. *BRAF* mutation was associated with increased sensitivity to inhibition of its downstream effectors MEK (*MAP2K1*) and ERK (*MAPK1*), and increased resistance to inhibition of the alternative RAF isoform gene CRAF (*RAF1*) and the MAPK regulators *PTPN11* and *SHOC2* (*Figure 4A and B*; *Hill et al., 2019*). *BRAF* mutation was also associated with increased sensitivity to inhibition of *PEA15,* presumably a result of the requirement of PEA15 for ERK dimerisation and signalling activity (*Formstecher et al., 2001*; *Herrero et al., 2015*).

Mutation or deletion of the tumour suppressor *RB1* (Retinoblastoma 1, Rb) was associated with increased sensitivity or resistance to inhibition of multiple Rb pathway members (*Figure 4C and D*). We found that *RB1* loss was reproducibly associated with resistance to inhibition of its negative regulators *CDK4* and *CDK6*, consistent both with the known Rb pathway structure and with preclinical data suggesting that *RB1* mutation confers resistance to CDK4/6 inhibitors (*Asghar et al., 2015*; *O'Leary et al., 2018*). Rb is a negative regulator of multiple E2F transcription factors, and we found that *RB1* loss was reproducibly associated with increased sensitivity to both *E2F1* and *E2F3* inhibition (*Figure 4C and D*). *RB1* loss was also associated with robust sensitivity to *SKP2*, a binding partner of Rb (*Ji et al., 2004*) first identified as an *RB1* synthetic lethal partner in retinoblastoma (*Xu et al., 2014*) and more recently as a highly penetrant *RB1* synthetic lethal partner in triple negative breast cancer (*Brough et al., 2018*; *Figure 4C and D*). Finally, *RB1* loss was reproducibly associated with increased sensitivity to inhibition of Cyclin Dependent Kinase 2 (*CDK2*), suggesting that it may be a useful biomarker for CDK2-specific inhibitors (*Tadesse et al., 2019*).

To test if this enrichment of pathway members among the robust dependencies associated with specific driver genes was a common phenomenon, for each driver gene with at least three dependencies we asked if these dependencies were enriched in specific signalling pathways (see Materials and methods). Of the twelve driver genes tested, we found that five of these were enriched in specific pathways and in all five cases found that the driver gene itself was also annotated as a member of the most enriched pathway (*Supplementary file 4*). As expected *RB*1 (most enriched pathway 'G1 Phase') and *BRAF* (most enriched pathway 'Negative feedback regulation of MAPK pathway') were among the five driver genes, alongside *PTEN* ('PI3K/AKT activation'), *CDKN2A* ('Cell cycle'), and *NRAS* ('Ras signaling pathway').

## Robust genetic interactions are enriched in protein–protein interaction pairs

In seeking to understand what particular characteristics robust genetic interactions might have, we noted that many of the robust genetic interactions we identified involved gene pairs whose protein products operate in the same pathway (e.g. the Rb pathway) or protein complex (e.g. SWI/SNF) suggesting that genetic interactions between gene pairs whose protein products physically interact may be more robust than other genetic interactions. To test this hypothesis, we compared the robust genetic interactions we identified with protein–protein interactions from the STRING protein–protein interaction database (*Szklarczyk et al., 2015*). We found that, when considering the set of all gene pairs tested, gene pairs identified as significant genetic interactors in at least one dataset are more likely to encode proteins that physically interact (*Figure 5A*) (Odds Ratio (OR) = 4.0, $p<2\times10^{-16}$, Fisher's Exact test). Furthermore, of the genetic interactions identified as significant in at least one dataset, those that are supported by a protein–protein interaction were significantly more likely to be reproduced in a second dataset (*Figure 5A*) (OR = 3.9, $p<1\times10^{-13}$). We therefore concluded

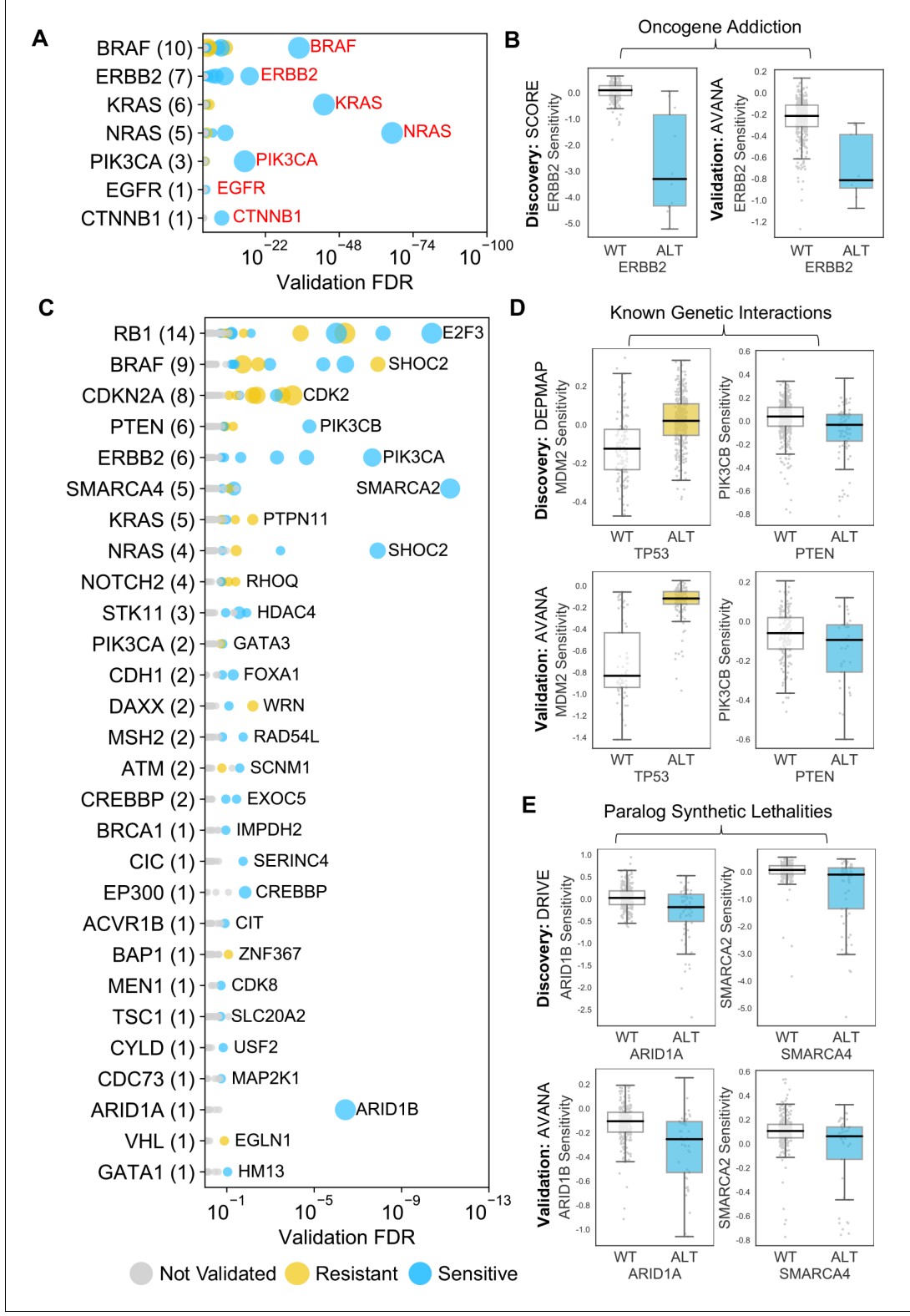

**Figure 3.** Identified robust genetic interactions. (**A**) Dot plot showing the robust genetic dependencies identified for oncogenes. Each coloured circle indicates a robust genetic dependency, scaled according to the number of dataset pairs it was validated in. The most significant genetic dependency (lowest FDR in a validation set) for each driver gene is labelled. Oncogenes are sorted by the number of robust dependencies and the total number of robust genetic dependencies for each driver gene is shown in parentheses. (**B**) Example of a validated oncogene

*Figure 3 continued on next page*

*Figure 3 continued*

addiction – *ERBB2* amplified cells are sensitive to *ERBB2* inhibition. Left shows the discovery dataset (SCORE) and right shows the validation dataset (AVANA). (**C**) Dot plot showing the robust genetic interactions identified for all driver genes. Each coloured circle indicates a robust genetic interaction, scaled according to the number of dataset pairs it was validated in. The most significant genetic interaction (lowest FDR in a validation set) for each driver gene is labelled. Drivers are sorted by the number of robust interactions and the total number of robust genetic interactions for each driver gene is shown in parentheses. *TP53* (132 robust genetic interactions) has been excluded for clarity, as have all self-self dependencies. (**D**) Examples of known genetic interactions identified from the integrated analysis, including an increased sensitivity of *PTEN* mutant tumour cell lines to *PIK3CB* inhibition and increased resistance of *TP53* mutant tumour cell lines to *MDM2* inhibition. Top row shows the data used to discover the interactions (DEPMAP dataset) while the bottom row shows the data used to validate the interactions (AVANA dataset with cell lines from DEPMAP excluded). (**E**) Synthetic lethal interactions involving paralog pairs. Top row shows the data used to discover the interactions (DRIVE dataset) while the bottom row shows the data used to validate the interactions (AVANA dataset with cell lines from DRIVE excluded).

The online version of this article includes the following figure supplement(s) for figure 3:

**Figure supplement 1.** Reproducible genetic dependencies include oncogene addictions.

that protein–protein interaction pairs are more likely to be significant hits in one dataset and even more likely to be reproduced across multiple datasets, suggesting this might be a feature of robust genetic interaction effects.

We noted that a large number (n = 132) of robust genetic interactions involved *TP53*, presumably as a result of the high number of *TP53* mutant tumour cell lines in the datasets (and its high mutation frequency in human cancer) and the associated increased statistical power to detect *TP53*-related genetic interactions. We therefore considered whether the significant number of *TP53*-related genetic interactions in our dataset could confound our analyses, especially as *TP53* is also associated with a disproportionately high number of protein interactions (>1700 medium confidence interactors in the STRING database alone, compared to a median of 37 medium confidence interactions across all proteins). However, even after excluding genetic interactions involving *TP53,* the observation that robust genetic interactions were enriched in protein–protein interaction pairs was still evident (*Figure 5B*); known protein interaction pairs were more likely to be identified as significant genetic interactions in one screen (OR = 3.8, p<$2\times10^{-16}$) and among the significant genetic interactions discovered in one screen, those involving protein–protein interaction pairs were more likely to be reproduced in a validation screen (OR = 9.3, p<$2\times10^{-16}$). The same effects were observed when considering genetic interactions observed at different false-discovery rate (FDR) thresholds (*Figure 5—figure supplement 1A,B*) and using different sources of protein–protein interaction data (*Figure 5—figure supplement 1C,D,E*, *Supplementary file 5*; *Alanis-Lobato et al., 2017*; *Chatr-Aryamontri et al., 2015*; *Huttlin et al., 2020*).

Previous work has demonstrated that the protein–protein interaction networks aggregated in databases are subject to significant ascertainment bias – some genes are more widely studied than others and this can result in them having more reported protein–protein interaction partners than other genes (*Rolland et al., 2014*). As cancer driver genes are studied more widely than most genes, they may be especially subject to this bias. To ensure the observed enrichment of protein–protein interactions among genetically interacting pairs was not simply due to this ascertainment bias, we compared the results observed for the real STRING protein–protein interaction network with 100 degree-matched randomised networks and again found that there was a higher than expected overlap between protein–protein interactions and both discovered and validated genetic interactions (*Figure 5—figure supplement 2*).

The increased reproducibility of genetic interactions associated with protein–protein interactions across different genetic perturbation screen datasets could have two distinct causes - increased reproducibility across distinct technologies or libraries (e.g. CRISPR/shRNA) or increased reproducibility/robustness of genetic interactions in cell line panels with distinct molecular backgrounds. To test the former possibility, we repeated our discovery/validation approach but focused on the set of cell lines that were common to different genetic perturbation screen datasets. Using this approach, the molecular backgrounds (i.e. cell lines) tested were the same, but the screening approach or library used differed. Upon doing this, we found that genetic interactions between gene pairs whose

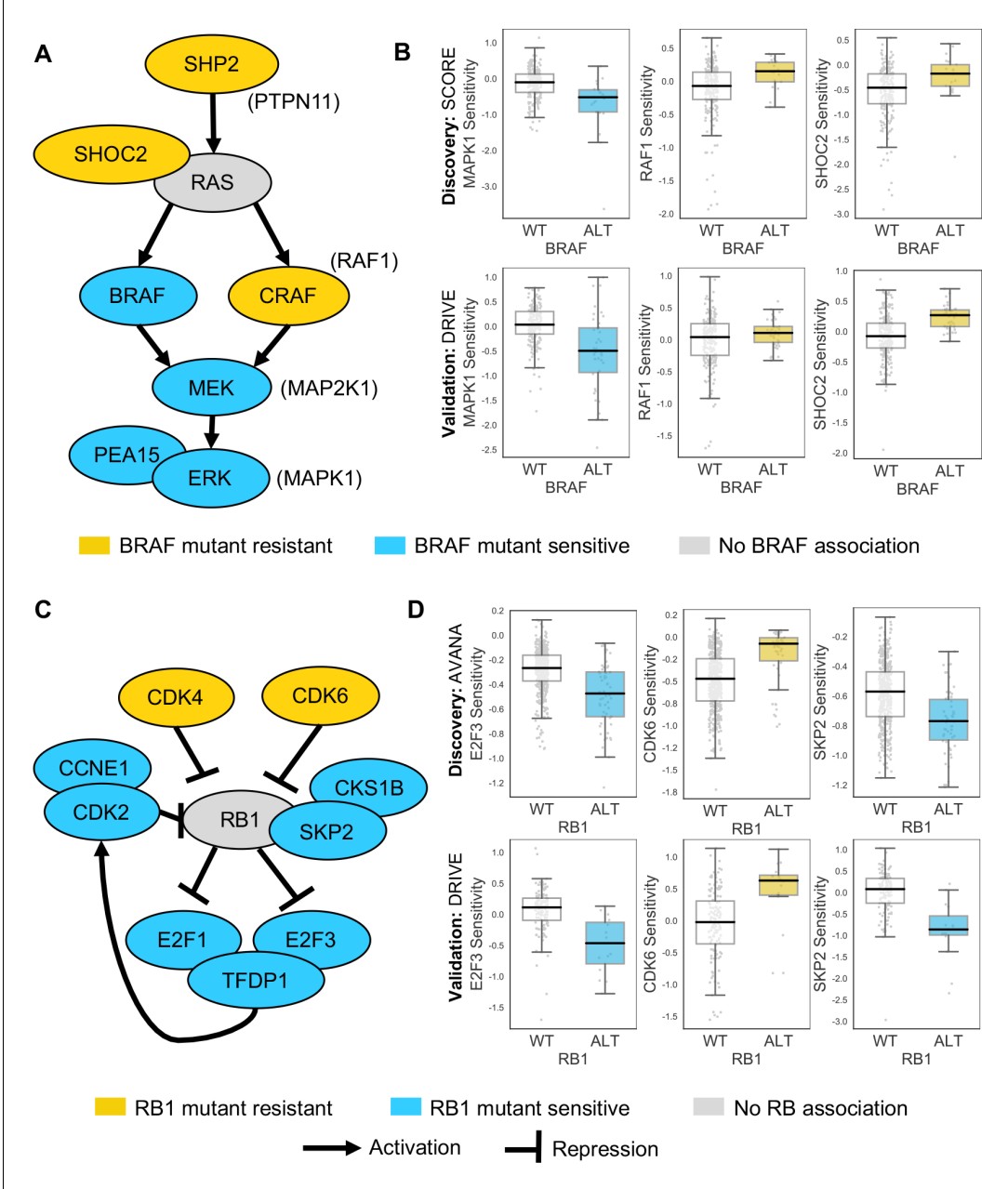

**Figure 4.** Robust genetic interactions involving *RB1* and *BRAF* recapitulate pathway relationships. (**A**) Simplified RAS/RAF/MEK/ERK pathway diagram. Protein names (e.g. MEK) are shown inside nodes, while associated gene names are shown adjacent (e.g. *MAP2K1*). Nodes are coloured according to their association with BRAF mutation - blue indicates increased sensitivity of *BRAF* mutant cell lines, yellow indicates increased resistance. (**B**) Boxplots showing selected genetic interactions associated with *BRAF* mutation. (**C**) Simplified Rb pathway diagram, highlighting robust genetic interactions involved in the Rb pathway. (**D**) Boxplots showing selected genetic interactions associated with *RB1* alteration.

protein products physically interact were significantly more reproducible across studies (*Figure 5C*, OR = 6.1 and p<2×10⁻¹⁰ when compared to discovered genetic interactions) (*Supplementary file 5*). To test reproducibility using the same screening approach across molecularly distinct cell lines, we artificially split individual datasets into non-overlapping discovery and validation sets of cell lines. Again, we found that genetic interactions between gene pairs whose protein products physically interact were more reproducible across distinct cell line panels (*Figure 5D*, OR = 8.0 and

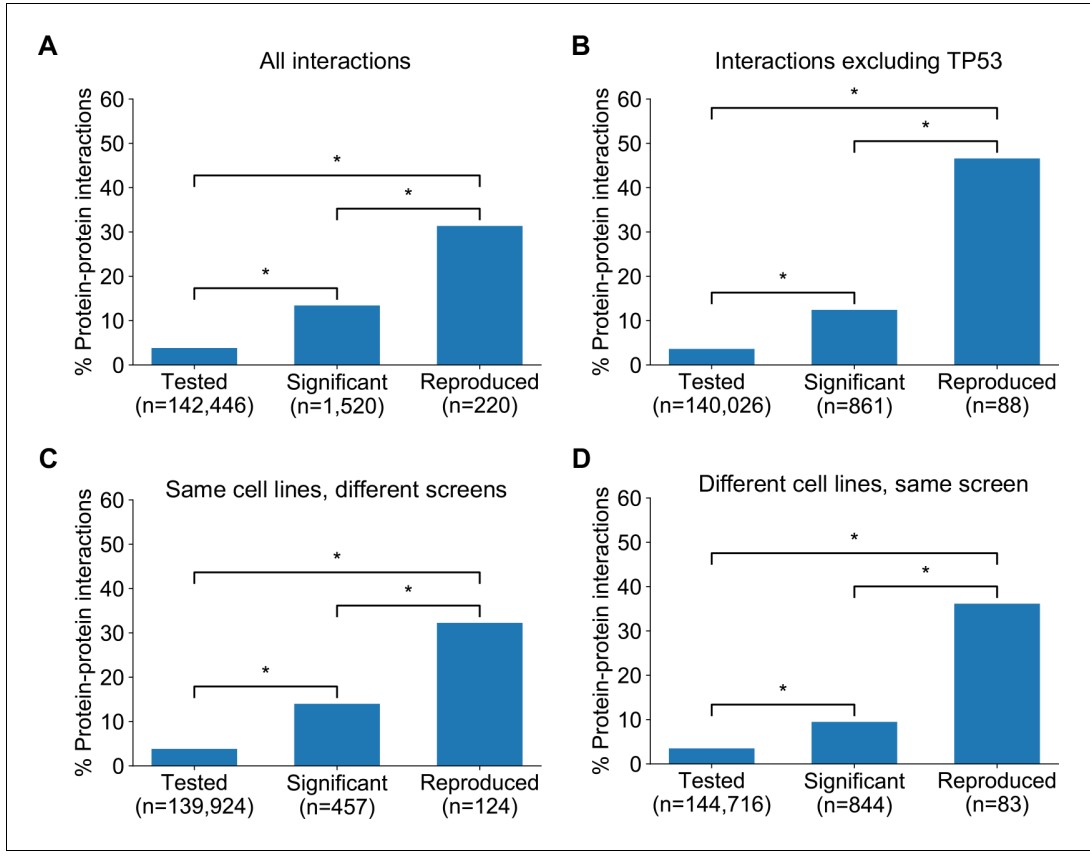

**Figure 5.** Robust genetic interactions are enriched in protein–protein interaction pairs. (A) Barchart showing the percentage of protein–protein interacting pairs observed among different groups of gene pairs. The groups represent all gene pairs tested, gene pairs found to be significantly interacting in at least one screen (FDR < 20%), and gene pairs found to reproducibly interact across multiple screens (i.e. a discovery and validation screen). Stars (*) indicate significant differences between groups, all significant at p<0.001 using Fisher's Exact Test. Odds ratios and p-values are provided in *Supplementary file 5*. (B) As A but with interactions associated with *TP53* removed. (C) As B but here the discovery and validation sets contain the same cell lines screened in different studies (e.g. 'AVANA ∩ DEPMAP' as discovery and 'DEPMAP ∩ AVANA' as validation). Consequently, reproducibility here means 'technical reproducibility' using different screening platforms. (D) Similar to B but here the discovery and validation sets contain single datasets partitioned into non-overlapping cell line sets (e.g. 'AVANA \ DEPMAP' as discovery and 'AVANA ∩ DEPMAP' as validation). Consequently, reproducibility here means 'genetic robustness' - the same association between gene pairs is observed across distinct genetic backgrounds.

The online version of this article includes the following figure supplement(s) for figure 5:

**Figure supplement 1.** Robust genetic interactions are enriched in protein–protein interaction pairs at different thresholds and using different databases.

**Figure supplement 2.** Genetic interactions are more enriched in real protein–protein interaction networks than randomised networks.

---

$p<1\times10^{-12}$ when compared to discovered genetic interactions) (*Supplementary file 5*). We therefore concluded that genetic interactions supported by protein–protein interactions were more reproducible across different screening approaches and across distinct cell line contexts, suggesting that these interactions are, overall, more robust.

## Prioritising robust synthetic lethal interactions from chemogenetic screens

As an alternative to genetic perturbation screening in large cell line panels, genetic interactions can also be identified using chemogenetic screens, where loss-of-function screens are performed in the presence and absence of specific small molecule inhibitors whose targets are relatively well defined.

Based on the observations made earlier, we hypothesised that genetic interactions identified in chemogenetic screens that involved genes whose protein products physically interact with the target of the inhibitor should both be more likely to be identified as genetic interaction partners in one screen and also more likely to be reproduced across multiple screens (i.e. to be more robust). To test this hypothesis, we analysed the results of a recent chemogenetic screen performed to identify genes whose loss is synthetic lethal with ATR inhibition (*Wang et al., 2019*). In this study, genome-wide CRISPR-Cas9 screens in three cell lines from different histologies (breast, kidney, colon) were used to identify genes whose inhibition is selectively essential in the presence of a small molecule ATR kinase inhibitor (*Vendetti et al., 2015*; *Wang et al., 2019*; *Figure 6A*).

As predicted, we found that protein interaction partners of ATR are more likely than random genes to be identified as a significant synthetic lethal interactor of ATR in at least one cell line

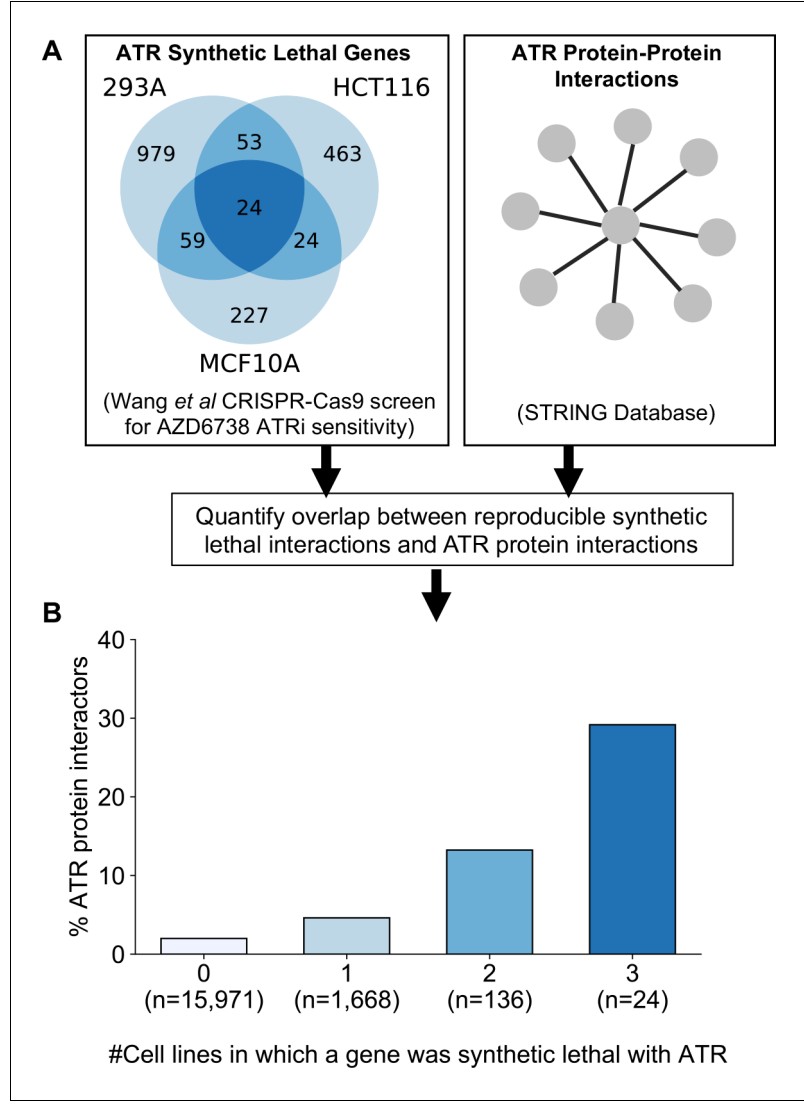

**Figure 6.** Reproducible ATR synthetic lethal interactions are enriched in ATR protein–protein interaction partners. (A) Workflow - synthetic lethal interactions from CRISPR-Cas9 screens in three cell lines (*Wang et al., 2019*) were compared to identify reproducible synthetic lethal partners. These genes were then compared with known ATR protein–protein interaction partners from the STRING database. (B) Bar chart showing the percentage of ATR protein interaction partners observed in different groups of genes. Genes are grouped according to whether they were identified as an ATR synthetic lethal partner in 0, 1, 2, or 3 cell line screens. Comparisons between all pairs of groups are significant at p<0.001 (Fisher's exact test) except for the comparison between genes that were hits in 2 and 3 cell lines (p=0.06).

(*Figure 6B*). Furthermore, we found that among the synthetic lethal interactions identified in at least one cell line, those involving known ATR protein interaction partners were significantly more likely to be reproduced in a second or even third cell line (*Figure 6B*). This suggests that, of the candidate genes identified in one screen, those that encode protein–protein interaction partners of ATR are significantly more likely to validate in additional contexts than genes with no known functional relationship to ATR.

## Prioritising robust synthetic lethal interactions involving passenger gene alterations

In addition to alterations of cancer driver genes, tumour cells typically harbour genetic alterations of large numbers of 'passenger' genes. Although these genes may not facilitate tumourigenesis or promote cancer cell growth, their alteration may still impart genetic vulnerabilities upon tumour cells. Indeed, multiple synthetic lethal interactions have been identified involving passenger genes that exhibit recurrent copy number loss in cancer cells due to their chromosomal proximity to tumour suppressor genes lost via loss-of-heterozygosity (*Kryukov et al., 2016*; *Marjon et al., 2016*; *Mavrakis et al., 2016*; *Muller et al., 2012*; *Muller et al., 2015*). The space of genetic interactions to test involving passenger gene alterations is much larger than that involving driver genes, as nearly every gene in the genome is either mutated or deleted in some cancer context. In addition, passenger genes are typically altered at frequencies lower than for driver genes and therefore the statistical power to identify genetic interactions associated with their alteration is somewhat reduced. With these issues in mind, we reasoned that protein–protein interaction maps might help narrow the search space considerably and thus reduce the burden of multiple hypothesis testing. For all passenger genes that were recurrently lost in at least ten tumour cell lines, either through homozygous deletion or loss-of-function mutation, we searched for genetic interactions with their protein–protein interaction partners using the same discovery and validation approach previously used for driver genes. In total we tested 47,781 interactions involving 2,972 passenger genes and 2149 selectively lethal genes. To perform an all-against-all test without filtering based on protein–protein interactions would have required more than six million tests, significantly increasing the burden of multiple-hypothesis testing. At an FDR of 20% we found 11 robust genetic interactions involving passenger gene alterations (*Supplementary file 7*). To assess whether this is more than would be expected by chance we randomly sampled 47,781 gene pairs from the same search space 100 times. The median number of robust genetic interactions identified amongst these randomly sampled gene pairs was 1 (mean 1.27, min 0, max 6) suggesting that the 11 robust genetic interactions observed among protein–protein interacting pairs was more than would be expected by chance. Three of the robust interactions involve genes frequently deleted with the tumour suppressor *CDKN2A* (*CDKN2B* and *MTAP*) and recapitulate known associations with *CDKN2A*. A further two genetic interactions involve a single chromosomal region (19p21.3) containing two interferon genes (*IFNB1* and *IFNW1*) which are frequently deleted together and consequently these two interactions really represent a single association (an increased sensitivity to thrombopoietin receptor *MPL*). Of the six remaining genetic interactions identified, four represent examples of paralog lethalities – loss of one member of a paralog pair is associated with increased sensitivity to the inhibition of the other member. *RPL22* loss was associated with increased sensitivity to its paralog *RPL22L1*, *TIMM17B* with its paralog *TIMM17A*, *DDX17* with its paralog *DDX5*, and *VPS4B* with its paralog *VPS4A*. We selected two of these robust synthetic lethal interactions for further validation – *VPS4B/VPS4A* and *DDX17/DDX5*.

VPS4A and VPS4B are highly sequence similar whole genome duplicates with protein sequence identity of 81%. Both proteins can form a complex with the Vacuolar protein sorting-associated protein VTA1 (*Huttlin et al., 2017*) and are involved in endosomal trafficking. *VPS4B* is located at 18q21 and is frequently deleted with the tumour suppressor *SMAD4*, explaining the relatively high frequency of loss of *VPS4B* in cancer. Previous analysis of the DRIVE shRNA dataset identified an association between *VPS4B* copy number loss and *VPS4A* sensitivity (*McDonald et al., 2017*). Here we find evidence of this association in two additional datasets – AVANA and SCORE (*Figure 7A*). Although this association is robust, it does not establish a causal link between *VPS4B* loss and *VPS4A* sensitivity. Indeed, there are 39 protein-coding genes on chromosome 18 located between *SMAD4* and *VPS4B*, any one of which could cause sensitivity to *VPS4A* inhibition. To verify that *VPS4B* is the cause of *VPS4A* sensitivity we transfected isogenic knockouts of either *VPS4A* and *VPS4B* with siRNA smartpools targeting either *VPS4A* or *VPS4B* and found that, consistent with a

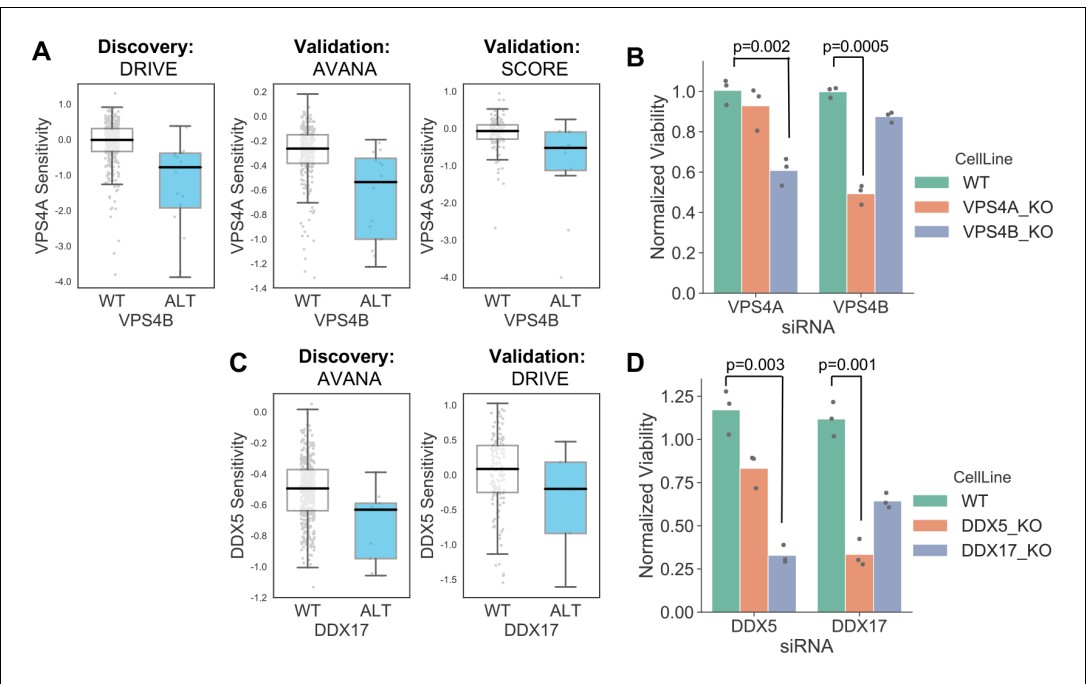

**Figure 7.** Robust synthetic lethalities associated with passenger gene loss. (**A**) Boxplots showing the association between *VPS4B* loss and *VPS4A* sensitivity in the discovery dataset (DRIVE) and two validation datasets (AVANA and SCORE). (**B**) Mean viability of HAP1 cells treated with siRNA smartpools targeting *VPS4A* or *VPS4B*. Individual data points are shown as black dots. Data are normalized within each cell line such that the mean viability of cells treated with a negative control (non-targeting scrambled siRNA) is equal to one and the mean viability treated with a positive control (siRNA smartpool targeting the broadly essential *PLK1* gene) is equal to 0. P-values from two-sided heteroscedastic T-tests. (**C**) Boxplots showing the association between *DDX17* loss and *DDX5* sensitivity in the discovery dataset (AVANA) and the validation dataset (DRIVE). (**D**) Mean viability of HAP1 cells treated with siRNA smartpools targeting *DDX5* or *DDX17*, normalization and statistics as per B.

negative genetic interaction between the two genes, compared to wildtype parental cells *VPS4A* knockout cells were more sensitive to siRNA targeting *VPS4B* and *VPS4B* knockout cells were more sensitive to siRNA targeting *VPS4A* (*Figure 7B*, *Supplementary file 8*).

Like VPS4A and VPS4B, DDX5 and DDX17 are widely conserved highly sequence similar whole genome duplicates (protein sequence identity 69%). They are DEAD box family RNA helicases that have multiple roles in both transcription and splicing; they act as coregulators for multiple transcription factors and also function as components of the spliceosome (*Dardenne et al., 2014*; *Fuller-Pace, 2013*). A direct protein–protein interaction between the two genes has also been reported (*Hegele et al., 2012*; *Huttlin et al., 2017*). *DDX17* is located at 22q12 in close proximity to the tumour suppressor *MYH9*, potentially explaining its recurrent deletion in tumour cell lines. We identified an association between *DDX17* loss and *DDX5* sensitivity in the AVANA CRISPR dataset and validated this association in the DRIVE shRNA dataset (*Figure 7C*). As with VPS4A/VPS4B, to verify that *DDX17 loss* is the cause of *DDX5* sensitivity we transfected isogenic knockouts of either *DDX17* and *DDX5* with siRNA smartpools targeting either *DDX5* or *DDX17*. We found that, consistent with a negative genetic interaction between the two genes, compared to wildtype parental cells *DDX17* knockout cells were more sensitive to siRNA targeting *DDX5* and *DDX5* knockout cells were more sensitive to siRNA targeting *DDX17* (*Figure 7D*, *Supplementary file 8*).

## Discussion

While the reproducibility of pharmacogenomic screens in cancer cell lines has been much discussed (*Cancer Cell Line Encyclopedia Consortium and Genomics of Drug Sensitivity in Cancer Consortium, 2015*; *Haibe-Kains et al., 2013*; *Niepel et al., 2019*), relatively little attention has been paid to the reproducibility of results from large-scale genetic screens in cell lines. Analyses of

the pharmacogenomic screen datasets have primarily focused on reproducibility in a very strict sense - that is quantifying the extent to which the same drug elicits the same response in the same cell line when assayed across different sites (*Niepel et al., 2019*). In some cases these analyses have been extended to quantify the extent to which the same associations between biomarkers and drugs can be observed across the same cell line panels assayed in different experiments (*Cancer Cell Line Encyclopedia Consortium and Genomics of Drug Sensitivity in Cancer Consortium, 2015*). Here we were interested in reproducibility in a much broader sense and sought to identify genetic interactions that could be reproduced both across distinct experiments and across distinct cell line panels, that is interactions that are robust to genetic and molecular heterogeneity. We developed an approach to identify these robust genetic interactions and used it to identify a set of 220 robust genetic interactions associated with cancer driver genes. We found that these robust genetic interactions are enriched among gene pairs whose protein products physically interact, suggesting a means by which we might prioritise the most promising candidates from screens for follow on studies.

We do not claim that our set of robust genetic interactions is comprehensive, as there are many reasons why real robust genetic interactions may not be identified by our approach. There are many driver genes that we have not included in our analysis because they are infrequently mutated in the datasets studied. Consequently, we can report no interactions for these genes. We have also focussed only on identifying interactions associated with mutation or copy number changes. There are likely to be dependencies associated with altered gene/protein expression that will be missed by this approach. Furthermore, for the genes that we do analyse, it is likely that some real interactions are not detected due to a lack of statistical power. Finally, of the dependencies identified in a discovery screen but absent in a validation screen, false negatives due to reagents with poor gene targeting ability likely play a significant role (*Kaelin, 2012*).

We have exclusively focussed on identifying dependencies that are evident across panels of cell lines from multiple cancer types ('pan-cancer dependencies'). It is likely that there are robust dependencies only evident *within* specific cancer types, but it is difficult to use our approach to identify them due to the restricted number of cell lines available for each cancer type. Even with a relatively common mutation (e.g. *KRAS* mutation in non-small cell lung cancer) it is challenging to partition the available cell lines into distinct discovery and validation sets while maintaining statistical power to identify potential dependencies. This issue may be alleviated by efforts to create large numbers of new tumour cell lines (*Boehm and Golub, 2015*) or through using isogenic models for discovery and cell line panels for validation (*Ryan et al., 2018*).

Many published synthetic lethal screens have focussed on identifying new drug targets for 'undruggable' oncogenes such as *MYC* and *RAS* (reviewed in *Cermelli et al., 2014* and *Downward, 2015* respectively). The rationale for such studies is that the oncogene addiction itself cannot be exploited directly and consequently a synthetic lethal approach is needed. However, here we found that for all oncogenes studied the most significant reproducible dependency identified was an oncogene addiction (*Figure 3A*). This suggests that any synthetic lethal interactions that are identified for oncogenes will likely be of a smaller effect size or operate in a more restricted context than the oncogene addiction itself. Previous work has suggested this to be true of *KRAS* (*McDonald et al., 2017*) but here we find that it appears to be a general property of all oncogenes studied. This implies that wherever possible, direct targeting of oncogenes might be more therapeutically effective than exploiting oncogene-related synthetic lethal effects.

Our approach to identify robust genetic dependencies involving cancer driver genes is unbiased in the sense that we did not incorporate prior knowledge of functional relationships to identify candidate gene pairs to test. Nonetheless, many of the robust synthetic lethalities identified reflect known biology. In particular, for each of the well-studied tumour suppressors *ARID1A, SMARCA4* and *PTEN* the most significant robust synthetic lethal interaction we identified has previously been reported in the literature. For *ARID1A*, its known synthetic lethal partner *ARID1B* was the only robust candidate interaction identified while for *PTEN* and *SMARCA4* their established synthetic lethal partners (*PIK3CB* and *SMARCA2* respectively) are the most significant robust hits by a large margin (*Figure 3C*). As with oncogenes, this suggests that if novel single gene vulnerabilities for these drivers are to be discovered, they may have a smaller effect size or operate in a more restricted setting.

In budding and fission yeast, multiple studies have shown that genetic interactions are enriched among protein–protein interaction pairs and *vice-versa* (*Costanzo et al., 2010*; *Kelley and Ideker, 2005*; *Michaut et al., 2011*; *Roguev et al., 2008*). Pairwise genetic interaction screens in individual

mammalian cell lines have also revealed an enrichment of genetic interactions among protein–protein interaction pairs (*Han et al., 2017*; *Roguev et al., 2013*). Our observation that discovered genetic interactions are enriched in protein–protein interaction pairs is consistent with these studies. However, these studies have not revealed what factors influence the conservation of genetic interactions across distinct genetic backgrounds, that is what predicts the robustness of a genetic interaction. In yeast, the genetic interaction mapping approach relies on mating gene deletion mutants and consequently the vast majority of reported genetic interactions are observed in a single genetic background (*Tong et al., 2001*). In mammalian cells, pairwise genetic interaction screens across multiple cell lines have revealed differences across cell lines but not identified what factors influence the conservation of genetic interactions across cell lines (*Shen et al., 2017*). While variation of genetic interactions across different strains or different genetic backgrounds has been poorly studied, previous work has analysed the conservation of genetic interactions across *species* and shown that genetic interactions between gene pairs whose protein products physically interact are more highly conserved (*Roguev et al., 2008*; *Ryan et al., 2012*; *Srivas et al., 2016*). Our analysis here suggests that the same principles may be used to identify genetic interactions conserved across genetically heterogeneous tumour cell lines. Previous work has also shown that genetic interactions between gene pairs involved in the same biological process, as indicated by annotation to the same gene ontology term, are more highly conserved across species (*Ryan et al., 2012*; *Srivas et al., 2016*). Similarly, genetic interactions that are stable across experimental conditions (e.g. in the presence and absence of different DNA damaging agents) are more likely to be conserved across species (*Srivas et al., 2016*). Although we have not tested them here, these additional features predictive of between-species conservation may also be predictive of robustness to genetic heterogeneity. Our set of robust genetic interactions may serve as the starting point for such analyses and may also serve as a training set for computational approaches to predict synthetic lethality (*Jerby-Arnon et al., 2014*).

Our finding that the robust genetic interactions associated with some driver genes can be interpreted in terms of the signalling pathway that the driver gene functions in suggests that pathway structure may also provide information on robustness. For example, it seems reasonable to hypothesise that synthetic lethal proteins that are in close vicinity in a pathway are more likely to exhibit a robust synthetic lethal than those that are more distantly connected. However, to allow such a hypothesis to be tested, we believe the annotation of molecular pathways should be somewhat more reliable and the set of experimentally-validated robust genetic interactions much larger.

Our results suggest that knowledge of protein–protein interactions could be used to improve the design and analysis of loss-of-function screens for synthetic lethal interactions. We have demonstrated the utility of incorporating such prior knowledge for identifying robust synthetic lethal interactions from genome-wide chemogenetic screens. We have also demonstrated that protein–protein interactions can aid the identification of genetic interactions associated with passenger gene alterations, where statistical power is limited due to relatively infrequent alterations and the number of potential interactions to test is enormous. An alternative to these approaches, where knowledge of protein–protein interactions is used after the screen has already been performed, would be to screen target libraries for specific driver genes based on their known protein interaction partners. Regardless of the approach used to identify candidate synthetic lethal interactions in a large-scale screen, our results suggest that candidates supported by a protein–protein interaction should be prioritised for follow on study as they are more likely to be robust to the genetic heterogeneity observed in tumour cells.

## Materials and methods

All data analysis was performed using Python 3.7 (RRID:SCR_008394), Pandas 0.24 (*McKinney, 2011*) (RRID:SCR_018214) and StatsModels 0.9.0 (*Seabold and Perktold, 2010*) (RRID:SCR_016074).

### Loss of function screens

Different scoring systems have been developed for calculating 'gene level' sensitivity scores from loss-of-function screens performed with multiple gene targeting reagents per gene (i.e. shRNAs or gRNAs). For the analysis of all loss-of-function screens we used the original authors' own preferred approaches. CERES sensitivity scores (*Meyers et al., 2017*) for AVANA (release 18Q4) were

obtained from the DepMap portal (https://depmap.org/portal/download/) (RRID:SCR_017655), while DEMETER v2 gene sensitivity scores for the DEPMAP shRNA screen (*Tsherniak et al., 2017*) were obtained from the same resource. For the DEPMAP screen, some genes were only screened in a subset of cell lines and these were excluded from all analyses. Quantile normalized CRISPRcleaned (*Iorio et al., 2018*) depletion log fold changes for Project SCORE (*Behan et al., 2019*) were obtained from the Project SCORE database (https://score.depmap.sanger.ac.uk/). ATARIS (*Shao et al., 2013*) scores for the DRIVE dataset (*McDonald et al., 2017*) were obtained from the authors. 28 of the 398 cell lines screened in DRIVE had missing gene scores for ~25% of genes screened and these cell lines were excluded from further analysis. All screens were mapped to a common cell line name format (that followed by the Cancer Cell Line Encyclopaedia [*Barretina et al., 2012*]) using the Cell Model Passports resource where appropriate (*van der Meer et al., 2019*).

## Identifying selectively lethal genes

Similar to previous work (*McDonald et al., 2017*; *Tsherniak et al., 2017*), to reduce the burden of multiple hypothesis testing we focused our analysis on genes whose inhibition appeared to cause growth defects in subsets of the cancer cell lines screened. That is, rather than testing for associations with genes whose inhibition was always lethal or never lethal, we focused our analyses on genes that could be associated with distinct sensitive and resistant cell line cohorts. We first identified a set of 'selectively lethal' genes using the AVANA dataset (*Meyers et al., 2017*) - those with a gene sensitivity score $<-0.6$ in at least 10 cell lines but no more than 259 cell lines (half of the screened cell lines). We augmented this with a list of 65 'outlier genes' identified by the authors of the DRIVE study as having a skewed distribution suggesting distinct sensitive and resistant cohorts (*McDonald et al., 2017*). Finally from the combined list we removed genes known to be commonly essential in cancer cell lines (*Hart et al., 2017*). This resulted in a set of 2470 selectively lethal genes (*Supplementary file 2*) which were used for all association analyses.

## Identifying gene alterations from copy number and exome profiling

For all cell lines we obtained sequencing data (CCLE_DepMap_18q3_maf_20180718.txt) and copy number profiles (public_18Q3_gene_cn_v2.csv) from the DepMap portal. These datasets contain integrated genotyping data from both the Cancer Cell Line Encyclopedia and GDSC resources (*Barretina et al., 2012*; *Cancer Cell Line Encyclopedia Consortium and Genomics of Drug Sensitivity in Cancer Consortium, 2015*; *Iorio et al., 2016*). We used this to identify likely functional alterations in a panel of cancer driver genes (*Vogelstein et al., 2013*) restricting our analysis to those genes that were subject to targeted sequencing as part of the Cancer Cell Line Encyclopedia (*Barretina et al., 2012*).

For most oncogenes we considered the gene to be functionally altered if it contained a protein altering mutation at a residue that is recurrently altered in either the COSMIC database or the Cancer Genome Atlas. For a small number of oncogenes (*ERBB2, CCND1, MDM2, MDM4*) we considered them to be functionally altered only if they were amplified. For all tumour suppressors we considered all protein-coding mutations and homozygous deletions to be functional alterations. We used a threshold of $-1.28$ to call homozygous deletions. The matrix of functional alterations to driver genes is presented in *Supplementary file 1*.

To identify loss-of-function alterations to passenger genes, a similar pipeline was used. However the driver genes from *Vogelstein et al., 2013* were excluded from analysis and only clear loss-of-function alterations were considered to be functional. The matrix of gene loss in passenger genes is presented in *Supplementary file 6*.

## Identifying genetic dependencies in individual datasets

We wished to identify associations between driver gene mutations and gene sensitivity scores that were not confounded by tissue specific gene sensitivity effects (e.g. SOX10 sensitivity scores can be naively associated with *BRAF* mutational status because SOX10 is essential in melanoma cell lines and *BRAF* mutation is common in melanoma). Thus, we wished to model gene sensitivity after first accounting for tissue type. To this end, associations between individual driver genes and gene sensitivity scores were identified using an ANOVA model that incorporated both tissue type and

mutational status as covariates, similar to the method previously developed for identifying pharmacogenomic interactions in cancer cell line panels (*Cokelaer et al., 2018*; *Iorio et al., 2016*). As recent work (*Behan et al., 2019*) has highlighted that some dependencies (e.g. *WRN*) can be associated with microsatellite instability rather than individual driver genes, we also incorporated microsatellite instability (*Ghandi et al., 2019*) as a covariate in our model. The model had the form 'gene_X_sensitivity ~MSI_status + C(Tissue) + driver_gene_Y_status' and was used to test the association between each recurrently mutated driver gene Y and all gene sensitivity scores X assayed in a given dataset. Driver genes were included in this analysis if they were functionally altered in at least five cell lines in the dataset being analysed. Correction for multiple hypothesis testing was performed using the *Benjamini and Hochberg, 1995* false discovery rate.

## Identifying genetic dependencies common to multiple datasets

When comparing a pair of datasets, we used one dataset as a discovery dataset and a second as a validation set, as outlined in *Figure 1C*. The discovery analysis was limited to the set of interactions that could be tested in both datasets, that is associations between the set of sensitivity scores for genes screened in both studies and the set of driver genes recurrently altered in both studies. An initial set of genetic interactions was identified in the discovery dataset at a specific FDR threshold and these associations were then tested in the validation set. We considered interactions to be reproduced in the validation dataset if: (1) the FDR was less than the threshold; (2) the uncorrected p-value was <0.05 and; (3) the sign of the association (sensitivity/resistance) was the same in both discovery and validation set. A FDR of 0.2 was used for all analysis presented in the main text but additional FDR thresholds (0.1, 0.3) were tested to ensure that all findings were robust to the exact choice of FDR (*Figure 5—figure supplement 1*).

## Pathway enrichment of genetic dependencies

Pathway enrichment was assessed using gProfiler (*Raudvere et al., 2019*) with KEGG (*Kanehisa et al., 2017*) and Reactome (*Jassal et al., 2020*) as annotation databases and the selectively lethal genes as the background list.

## Protein–protein interactions

Protein–protein interactions were obtained from STRING v10.5 (*Szklarczyk et al., 2015*) (RRID:SCR_005223), BIOGRID 3.5.170 (*Chatr-Aryamontri et al., 2015*) (RRID:SCR_007393), HIPPIE v2.0 (*Alanis-Lobato et al., 2017*)(RRID:SCR_014651) and BioPlex 3.0 (*Huttlin et al., 2020*) (RRID:SCR_016144). Results in the main text make use of medium confidence STRING interactions (STRING integrated score >0.4). However, to ensure robustness to the thresholds shown, all analyses were repeated for the full set of HIPPIE, BioGRID and BioPlex interactions (*Figure 3—figure supplement 1*). NetworkX version 2.2 (*Hagberg et al., 2008*)(RRID:SCR_016864) was used to create 100 randomised versions of the medium confidence STRING interaction network analysed in *Figure 5—figure supplement 2*.

## siRNA experiments

HAP1 cell lines were obtained from Horizon Discovery: HAP1_WT (C631), HAP1_VPS4A_ KO (HZGHC004623c005), HAP1_VPS4B_KO (HZGHC006889c011), HAP1_DDX5_KO (HZGHC006136c012) and HAP1_DDX17_KO (HZGHC007221c009). Cells were cultured in IMDM (10–016-CV; Corning) supplemented with 10% FBS (10270–106; Thermo Fisher Scientific). ON-TARGETplus siRNA SMARTpools targeting *VPS4A* (L-013092-00-0005), *VPS4B* (L-013119-00-0005), *DDX5* (L-003774-00-0005), *DDX17* (L-013450-01-0005), *PLK1* (L-003290-00-0005) and a non-targeting scramble control (D-001810-10-20) were obtained from Dharmacon. HAP1 cells were seeded to a density of 5000 cells per well of a 96-well plate and 5 nM siRNA was transfected with Lipofectamine 3000 (L3000015; Thermo Fisher Scientific) in Opti-MEM I Reduced Serum Medium (31985070; Thermo Fisher Scientific). Cell viability was measured 72 hr after siRNA transfection using CellTiter-Glo Luminescent Cell Viability Assay (G7570; Promega). The 96 well plates were read using a SpectraMax M3 Microplate Reader (Molecular devices). Viability effects for each siRNA targeting each gene X in each cell line y were normalised using the following formula:

NormalisedViability $(siRNA\_X_y) = 1 - (siCTRL_y - siRNA\_X_y) / (siCTRL_y - siPLK1_y)$ where $siCTRL_y$ is the average of 3 measurements of non-targeting scramble control in cell line y and $siPLK1_y$ is the

average of 3 measurements of an siRNA smartpool targeting *PLK1* in cell line y. Raw and normalised viability data are in *Supplementary file 8*.

## Acknowledgements

CJL's work on this study was funded by Breast Cancer Now as part of Programme Funding to the Breast Cancer Now Toby Robins Research Centre (CTR-Q4-Y2) and via a Cancer Research UK Programme Grant (CRUK/A14276). CJR's work on this study was funded by the HRB/SFI/Wellcome Trust partnership (grant number 103049/Z/13/Z) and by the Irish Research Council Laureate Awards 2017/2018. We thank members of the Ryan and Lord labs for helpful comments and discussions.

## Additional information

### Funding

| Funder | Grant reference number | Author |
| --- | --- | --- |
| Irish Research Council | Laureate Awards 2017/2018 | Colm J Ryan |
| Cancer Research UK | CRUK/A14276 | Christopher J Lord |
| Breast Cancer Now | CTR-Q4-Y2 | Christopher J Lord |
| Wellcome Trust | SFI-HRB-Wellcome Trust Biomedical Research Partnership 103049/Z/13/Z | Colm J Ryan |
| Science Foundation Ireland | SFI-HRB-Wellcome Trust Biomedical Research Partnership 103049/Z/13/Z | Colm J Ryan |
| Health Research Board | SFI-HRB-Wellcome Trust Biomedical Research Partnership 103049/Z/13/Z | Colm J Ryan |

The funders had no role in study design, data collection and interpretation, or the decision to submit the work for publication.

### Author contributions

Christopher J Lord, Conceptualization, Funding acquisition, Writing - original draft, Writing - review and editing; Niall Quinn, Validation, Investigation; Colm J Ryan, Conceptualization, Data curation, Formal analysis, Supervision, Funding acquisition, Validation, Visualization, Methodology, Writing - original draft, Writing - review and editing

### Author ORCIDs

Colm J Ryan  https://orcid.org/0000-0003-2750-9854

### Decision letter and Author response

Decision letter https://doi.org/10.7554/eLife.58925.sa1
Author response https://doi.org/10.7554/eLife.58925.sa2

## Additional files

### Supplementary files

- Supplementary file 1. Driver gene alterations.
- Supplementary file 2. Selectively lethal genes.
- Supplementary file 3. Reproducible genetic dependencies.
- Supplementary file 4. Pathway enrichment of driver gene dependencies.
- Supplementary file 5. Protein–protein interaction enrichment.
- Supplementary file 6. Passenger gene loss alterations.

- Supplementary file 7. Passenger gene dependencies.
- Supplementary file 8. VPS4A_VPS4B and DDX5_DDX17 viability data.

### Data availability

All data generated during this study are included in the manuscript and supporting files.

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
