## [Decision Letter]

[Editors' note: this paper was reviewed by Review Commons.]

**Acceptance summary:**

In this manuscript the authors develop a computational approach designed to identify robust genetic interactions that can be used to predict tumor cell genetic vulnerabilities. The authors find that oncogene addiction, as opposed to synthetic lethality, tends to be a more robust predictor of genetic dependencies in tumor cells. They also find that robust genetic interactions in cancer are enriched for gene pairs whose protein products physically interact. Therefore, the latter could be considered a surrogate in target selection for tumors with currently undruggable driver oncogenes.

---

## [Author Response]

We thank both reviewers for their positive assessment of our manuscript and their insightful comments. In light of the reviews, we have not only strengthened the manuscript by the inclusion of new analyses that further support the main conclusions of the work but have also re-drafted the manuscript for clarity and to address the points made by the reviewers.

We have provided an updated version of our manuscript. In addition to the text changes we note that we have added:

- A new supplemental table (Supplementary file 4) showing the enrichment of dependencies associated with specific driver genes in signalling pathways.

- A new panel (Figure 5—figure supplement 1E) to one of the supplemental figures demonstrating that robust genetic interaction pairs are enriched in protein-protein interactions from the systematically generated BioPLEX protein-protein interaction network (recently described in Huttlin et al. bioRxiv 2020.01.19.905109).

- A new supplemental figure (Figure 5—figure supplement 2) demonstrating that the enrichment of STRING protein-protein interactions among both discovered and reproduced genetic interactions is more than that observed in degree preserving randomisations of the STRING network.

We provide a point-by-point response below.

Reviewer #1 (Evidence, reproducibility and clarity):Summary:Reproducibility of genetic interactions across studies is low. The authors identify reproducible genetic interactions and ask the question of “What are properties of robust genetic interactions?”. They find that 1. oncogene addiction tends to be more robust than synthetic lethality and 2. genetic interactions among physically interacting proteins tend to be more robust. They then use protein-protein interactions (PPIs) to guide the detection of genetic interactions involving passenger gene alterations.Major comments:The claims of the manuscript are clear and well supported by computational analyses. My only concern is the influence of (study) bias on the observed enrichment of physical protein interactions among genetic interactions. 1. Due to higher statistical power the here described approach favors genetic interactions involving frequently altered cancer genes (as acknowledged by the authors). 2. Also some of the libraries in the genetic screens might be biased towards better characterized screens. 3. PPI networks are highly biased towards well studied proteins (in which well-studied proteins – in particular cancer-related proteins – are more likely to interact). The following tests would help to clarify if and to which extend these biases contribute to the described observations:

We thank the reviewer for the positive assessment of our manuscript and have addressed the issue of study bias in response to the specific queries below.

1) The authors should demonstrate that the PPI enrichment in reproducible vs nonreproducible genetic interactions is not solely due to the biased nature of PPI networks. One simple way of doing so would be to do the same analysis with a PPI network derived from a single screen (e.g. PMID: 25416956). I assume that due to the much lower coverage the effect will be largely reduced but it would be reconfirming to see a similar trend in addition to the networks on which the authors are already testing. Another way would be to use a randomized network (with the same degree distribution as the networks the authors are using and then picking degree matched random nodes) in which the observed effect should vanish.

We appreciate the reviewer’s point and have now assessed both of the suggested approaches.

The overlap with unbiased yeast two-hybrid (y2h) screens, even the recent HuRI dataset (Luck et al., Nature 2020), was too small in scale to draw any conclusions. Among the ~140,000 interactions tested for genetic interactions, only 51 overlap with y2h interactions. Two of the discovered genetic interactions were supported by a y2h interaction, while one of the robust genetic interactions was supported by a y2h interaction. While this is actually more than would be expected based on the overlap of interactions in the test space the numbers are not especially convincing.

We therefore focused on two alternative assessments. We first compared our results with the network derived from the systematic AP-MS mapping of protein interactions in HEK293 cells (BioPlex 3.0, Huttlin et al., bioRxiv 2020). We restricted our analysis of genetic interactions to gene pairs that could conceivably be observed in the BioPlex dataset (i.e. between baits screened and preys expressed in HEK293T). We found that although the numbers were small, the same pattern of enrichment was observed:

This analysis has now been added to the revised manuscript as part of the existing Supplementary file 4 and as new figure panel Figure 5—figure supplement 1E.

We next compared the results we observed with the real STRING protein-protein interaction network to 100 degree-matched randomisations of this network. We observed that the number of discovered and validated genetic interactions observed using the real STRING interaction network was greater than that observed using the randomised networks. With this in mind, we have now revised the manuscript to state:

“Previous work has demonstrated that the protein-protein interaction networks aggregated in databases are subject to significant ascertainment bias – some genes are more widely studied than others and this can result in them having more reported protein-protein interaction partners than other genes (Rolland et al., 2014). As cancer driver genes are studied more widely than most genes, they may be especially subject to this bias. To ensure the observed enrichment of protein-protein interactions among genetically interacting pairs was not simply due to this ascertainment bias, we compared the results observed for the real STRING protein-protein interaction network with 100 degree-matched randomised networks and again found that there was a higher than expected overlap between protein-protein interactions and both discovered and validated genetic interactions (Figure 5—figure supplement 2).”

2) What's the expected number of robust genetic interactions involving passenger gene alterations? Is it surprising to identify 11 interactions? This question could be addressed with some sort of randomization test: When selecting (multiple times) 47,781 non-interacting random pairs between the 2,972 passenger genes and 2,149 selectively lethal genes, how many of those pairs form robust genetic interactions?

We have now addressed this as follows:

“At an FDR of 20% we found 11 robust genetic interactions involving passenger gene alterations (Supplementary file 6). To assess whether this is more than would be expected by chance we randomly sampled 47,781 gene pairs from the same search space 100 times. The median number of robust genetic interactions identified amongst these randomly sampled gene pairs was 1 (mean 1.27, min 0, max 6) suggesting that the 11 robust genetic interactions observed among protein-protein interacting pairs was more than would be expected by chance.”

Reviewer #1 (Significance):Personalized cancer medicine aims at the identification of patient-specific vulnerabilities which allow to target cancer cells in the context of a specific genotype. Many oncogenic mutations cannot be targeted with drugs directly. The identification of genetic interactions is therefore of crucial importance. Unfortunately, genetic interactions show little reproducibility across studies. The authors make an important contribution to understanding which factors contribute to this reproducibility and thereby providing means to also identify more reliable genetic interactions with high potential for clinical exploitation or involving passenger gene alterations (which are otherwise harder to detect for statistical reasons).Reviewer #2 (Evidence, reproducibility and clarity):In this manuscript, Lord et al. describe the analysis of loss-of-function (LOF) screens in cancer cell lines to identify robust (i.e., technically reproducible and shared across cell lines) genetic dependencies. The authors integrate data from 4 large-scale LOF studies (DRIVE, AVANA, DEPMAP and SCORE) to estimate the reproducibility of their individual findings and examine their agreement with other types of functional information, such as physical binding. The main conclusions from the analyses are that: a) oncogene-driven cancer cell lines are more sensitive to the inhibition of the oncogene itself than any other gene in the genome; b) robust genetic interactions (i.e., those observed in multiple datasets and cell lines driven by the same oncogene/tumour suppressor) are enriched for gene pairs encoding physically interacting proteins.Main comments:I think this study is well designed, rigorously conducted and clearly explained. The conclusions are consistent with the results and I don't have any major suggestions for improving their support. I do, however, have a few suggestions for clarifying the message.

We thank the reviewer for this positive assessment of our manuscript and have addressed the requests for clarity below.

Could the authors provide some intuitive explanation (or speculation) about the 2 observed cases of tumour suppressor "addiction” (TP53 and CDKN2A)? While the oncogene addiction cases are relatively easy to interpret, the same effects on tumour suppressors are less clear. Is it basically an epistatic effect, which looks like a relative disadvantage? For example, if we measure fitness: TP53-wt = 1, TP53-wt + CRISPR-TP53 = 1.5, TP53-mut = 1.5, TP53-mut + CRISPR-TP53 = 1.5. That is, inhibiting TP53 in TP53 mutant cells appears to be disadvantageous (relative to WT) only because inhibiting TP53 in wild-type cells is advantageous?

The reviewer is correct – the *TP53*/*TP53* dependency is similar to an epistatic effect. In a *TP53* mutant background targeting *TP53* with shRNA or CRISPR has a neutral effect, while in a TP53 wild type background targeting TP53 with shRNA or CRISPR often causes an increase in cell growth. We have clarified this in the text below:

“We also identified two (2/9) examples of “self *vs.* self” dependencies involving tumour suppressors – *TP53* (aka p53) and *CDKN2A* (aka p16/p14arf) (Figure 3—figure supplement 1C). […] A similar effect was observed for *CDKN2A,* although the growth increase resulting from inhibiting *CDKN2A* in wild-type cells is much lower than that seen for *TP53* (Figure 3—figure supplement 1C).”

In the analysis of overlap between genetic and physical interactions, the result should be presented more precisely. Currently, the text reads "when considering the set of all gene pairs tested, gene pairs whose protein products physically interact were more likely to be identified as significant genetic interactors". However, the referenced figure (Figure 5A) shows an orthogonal perspective: relative to all gene pairs tested, those that have a significant genetic interaction are more likely to have a physical interaction as well. In other words, in the text, we are comparing the relative abundance of genetic interactions in 2 sets: tested and physically interacting. However, in the figure, we are comparing the relative abundance of protein interactions in 2 sets – tested and genetically interacting. The odds ratio and the p-values stay the same, but the result would be clearer if the figure matched the description in the text.

Due to the fact that genetic interactions are rare (~1% of all gene pairs tested have a discovered genetic interaction, ~0.1% have a validated genetic interaction) it’s hard to convey the enrichment effectively. This is demonstrated in Author response image 1 – it’s clear that there are more discovered/validated genetic interaction pairs among the protein-protein interaction pairs, but the scale is hard to appreciate:

**Author response image 1. sa2fig1:** 

Focusing only on the discovered/validated genetic interactions makes the picture a little clearer but does not effectively show that the discovered pairs themselves are enriched among protein-protein interaction pairs.

As we feel the original figures convey the main message most effectively, we have altered the text rather than the images as follows:“We found that, when considering the set of all gene pairs tested, gene pairs identified as significant genetic interactors in at least one dataset are more likely to encode proteins that physically interact (Figure 5A).”

Reviewer #2 (Significance):The identification of a significant overlap between genetic and physical interactions in cancer cell lines is an interesting and promising observation that will help understanding known genetic dependencies and predicting new ones. However, similar observations have been made in other organisms and biological systems. These past studies should be referenced to provide a historical perspective and help define further analyses in the cancer context. In particular, studies in yeast *S. cerevisiae* have shown that, not only there is a general overlap between genetic interactions (both positive and negative) and physical interactions, but at least 2 additional features are informative about the relationship: a) the relative strength of genetic interactions and b) the relative density of physical interactions (i.e., isolated interaction vs protein complexes). Here's a sample of relevant studies: 1) von Mering et al., Nature, 2002; 2) Kelley & Ideker, 2005) Bandyopadhyay et al., PLOS Comput Biol, 2008; 4) Ulitsky et al., Mol Syst Biol, 2008; 5) Baryshnikova et al., Nat Methods, 2010; 6) Costanzo et al., 2010) Costanzo et al., Science, 2016.Similar observations have also been made in mammalian systems: e.g., in mouse fibroblasts (Roguev et al., 2013) and K562 leukemia cells (Han et al., 2017). I don't think that past observations negate the novelty of this manuscript. The analysis presented here is more focused and more comprehensive as it is based on a large integrated dataset and is driven by a series of specific hypotheses. However, a reference to previous publications should be made.As a frame of reference: my expertise is in high-throughput genetics of model organisms, including mapping and analyzing genetic interactions.

We thank the reviewer for highlighting this point.

We have attempted to provide better context for our work in the discussion as follows:

“In budding and fission yeast, multiple studies have shown that genetic interactions are enriched among protein-protein interaction pairs and vice-versa (Costanzo et al., 2010; Kelley and Ideker, 2005; Michaut et al., 2011; Roguev et al., 2008). […] Our analysis here suggests that the same principles may be used to identify genetic interactions conserved across genetically heterogeneous tumour cell lines.”